# SARS-CoV-2 antibody prevalence in a pediatric cohort of unvaccinated children in Mérida, Yucatán, México

**Guadalupe Ayora-Talavera**[1☯]**, Oscar D. Kirstein**[2☯]**, Henry Puerta-Guardo**[1,3]**, Gloria A. Barrera-Fuentes**[3,4]**, Desiree Ortegòn-Abud**[3,4]**, Azael Che-Mendoza**[3]**, Manuel Parra**[1,3]**, Fernando Peña-Miranda**[5]**, Carlos Culquichicon**[2,6]**, Norma Pavia-Ruz**[4]**, Afshin Beheshti**[7,8,9]**, Nídia S. Trovão**[8,10]**, Pilar Granja-Pérez**[5]**, Pablo Manrique-Saide**[3]**, Gonzalo M. Vazquez-Prokopec**[2]**, James T. Earnest**[2]*

**1** Virology Laboratory, Centro de Investigaciones Regionales Dr. Hideyo Noguchi, Universidad Autónoma de Yucatán, Mérida, Yucatán, México, **2** Department of Environmental Sciences, Emory University, Atlanta, GA, United States of America, **3** Campus de Ciencias Biológicas y Agropecuarias, Universidad Autónoma de Yucatán, Mérida, Yucatán, México, **4** Hematology Laboratory, Centro de Investigaciones Regionales Dr. Hideyo Noguchi, Universidad Autónoma de Yucatán, Mérida, Yucatán, México, **5** Laboratorio Estatal de Salud Publica, Mérida, Yucatán, México, **6** Rollins School of Public Health, Emory University, Atlanta, GA, United States of America, **7** KBR, Space Biosciences Division, NASA Ames Research Center, Moffett Field, CA, United States of America, **8** COVID-19 International Research Team, Medford, MA, United States of America, **9** Stanley Center for Psychiatric Research, Broad Institute of MIT and Harvard, Cambridge, MA, United States of America, **10** Division of International Epidemiology and Population Studies, Fogarty International Center, National Institutes of Health, Bethesda, Maryland, United States of America

☯ These authors contributed equally to this work.
* jtearne@emory.edu

**Data Availability Statement:** All data used for this report can be found at Mendeley data (https://data.mendeley.com/drafts/h43mdmz6vv).

## Abstract

The prevalence of SARS-CoV-2 exposure in children during the global COVID-19 pandemic has been underestimated due to lack of testing and the relatively mild symptoms in adolescents. Understanding the exposure rates in the pediatric population is essential as children are the last to receive vaccines and can act as a source for SARS-CoV-2 mutants that may threaten vaccine escape. This cross-sectional study aims to quantify the prevalence of anti-SARS-CoV-2 serum antibodies in children in a major city in México in the Spring of 2021 and determine if there are any demographic or socioeconomic correlating factors. We obtained socioeconomic information and blood samples from 1,005 children from 50 neighborhood clusters in Mérida, Yucatán, México. We then tested the sera of these participants for anti-SARS-CoV-2 IgG and IgM antibodies using lateral flow immunochromatography. We found that 25.5% of children in our cohort were positive for anti-SARS-CoV-2 antibodies and there was no correlation between age and antibody prevalence. Children that lived with large families were statistically more likely to have antibodies against SARS-CoV-2. Spatial analyses identified two hotspots of high SARS-CoV-2 seroprevalence in the west of the city. These results indicate that a large urban population of unvaccinated children has been exposed to SARS-CoV-2 and that a major correlating factor was the number of people within the child's household with a minor correlation with particular geographical hotspots. There is also a larger population of children that may be susceptible to future infection upon easing of social distancing measures. These findings suggest that in future pandemic scenarios,

**Funding:** This work was conducted using 'future use' blood specimens collected by the TIRS-project located in Mérida (NIH: U01AI148069) and receiving support from the Universidad Autónoma de Yucatán to G.V.P. . A.B. was supported by supplemental funds for COVID-19 research from Translational Research Institute of Space Health through NASA Cooperative Agreement NNX16AO69A (T-0404), and further funding from KBR, Inc. G.A.T. received funds from CONACYT grant 314296. The funders had no role in study design, data collection and analysis, decision to publish, or preparation of the manuscript.

limited public health resources can be best utilized on children living in large households in urban areas.

## Introduction

Countries in Latin America and the Caribbean have reported over 43 million cases of severe acute respiratory syndrome coronavirus 2 (SARS-CoV-2) by November of 2021 [1]. According to the World Health Organization, more than 3.5 million SARS-CoV-2 infections have been reported in México from December 2019 to September 2021, leading to over 250,000 reported deaths by the coronavirus disease 2019 (COVID-19) [2]. Even as effective vaccines against SARS-CoV-2 are distributed, there are public health threats due to low vaccination rates and virus variants.

Population prevalence of SARS-CoV-2 exposure is an important factor for understanding the effectiveness of control methods and predicting the susceptibility of a given population to future outbreaks. Lack of data on SARS-CoV-2 exposure in children has contributed to inaccurate population prevalence estimates [3]. This is due to several factors, such as the lack of testing supplies and access to a large ambulatory pediatric cohort and the fact that COVID-19 is generally mild in children. Since asymptomatic children may act as reservoirs for SARS-CoV-2, information about their immunity can help guide school reopening policies and vaccination plans in countries with limited resources for population-wide immunization [4,5].

This study was performed to provide insight on the prevalence of SARS-CoV-2 exposure in children in the city of Mérida, Yucatán, México. Mérida is the capital of Yucatán State and the largest city (population ~1 million) in the Yucatán Peninsula of México. The first case of SARS-CoV-2 in Yucatán, imported from Spain, was reported on March 12, 2020. Since then, there have been 44,386 reported infections and ~5,200 coronavirus-related deaths in Yucatán. Mérida accounted for 64.1% of all reported cases in the state. Overall, the progression of the COVID-19 outbreaks in Mérida has been similar to that observed in the most affected areas in Latin American countries, with two waves of infection peaking from June to August 2020 and May and September 2021 [6]. Of the 44,386 SARS-CoV-2 infections reported in Yucatán during 2020–2021, only 3% manifested in the pediatric population (≤15 years old).

We selected 1,005 banked serum samples collected from children in the spring of 2021 from over 50 neighborhood clusters located throughout the city of Mérida, Yuctán, México. Each serum sample was tested for the presence of either immunoglobulin G or M (IgG or IgM) antibodies specific for a recombinant SARS-CoV-2 spike protein. We then performed statistical analyses and geographical modeling to determine risk factors for previous SARS-CoV-2 exposure. Our results provide a unique perspective of SARS-CoV-2 exposure in a Latin American population and could inform public health and vaccination campaigns in future pandemics.

## Methods

### Statement of ethics

This study capitalized on a pediatric cohort established to evaluate the epidemiological impact of indoor residual spraying on arbovirus transmission and utilized serum stored after either participant written consent or verbal assent was attained [7]. The parents or guardians of all participants provided informed written consent for participation in this study. All participants agreed to have their specimens kept for future use. All procedures for participant assent/

consent, blood acquisition, and serum storage were approved by the Institutional Review Boards of Emory University (IRB00108666) and the Universidad Autónoma de Yucatán (UADY) (CEI-05-2020).

## Study site

The current study was conducted in Mérida, the capital city of Yucatán State, located in the southeast peninsula of México. Mérida is the largest urban center in the region, with 995,129 inhabitants (48.2% male and 51.8% female) in 2020 (https://datamexico.org). The city center of Mérida is located at 20˚ 58′ 12″ N and 89˚ 37′ 12″ W.

## Research design

During the COVID-19 outbreak, researchers from Emory University and UADY began recruiting a cohort of children aged 2–15 years old in October 2020 as a part of the Targeted Indoor Residual Spraying (TIRS) trial in Mérida [7]. The trial's primary endpoint is to quantify the epidemiological impact of the novel mosquito control strategy, Targeted Indoor Residual Spraying (TIRS), in arbovirus (ABV) infections [7]. The study enrolled 4,600 children distributed among 50 randomized clusters (5x5 city blocks) in Mérida. Due to the co-occurrence with SARS-CoV-2 infections and frequent reports of comorbidities with ABV diseases in the study population, we examined 1,005 serum banked samples of children participating in the TIRS trial for anti-SARS-CoV-2 antibody testing. These samples were all taken between February and June 2021 and tested between July and August of that year. The participants agreed to future use of their blood specimens and the consent forms have been made available (doi:10. 17632/h43mdmz6vv.1). The samples were selected randomly and with no previous information about exposure or infection with the SARS-CoV-2 virus. If more than one participating child lived in the same home, both participants' samples were tested for this study.

## Sampling procedures

Field personnel collected blood specimens by house-to-house visits following COVID-19 guidelines proposed by the Centers for Disease Control and Prevention (CDC) and the Yucatán Ministry of Health. Trained and certified nurses collected 5 ml of blood by venipuncture. Additionally, social workers conducted a socio-demographic survey to the head of each household. Collected blood samples were processed following the previously published TIRS protocol [7]. Briefly, venous blood was collected using the BD Vacutainer collection system with serum-separating tubes. For serology, tubes were transported to the Yucatán State Diagnostics Laboratory (Ministry of Health) and centrifuged at 13,000 rpm for 15 min; then, sera were aliquoted and stored at −80˚C at UADY.

## SARS-CoV-2 IgM/IgG antibody detection

Sero-reactivity of the collected serum samples was analyzed by a lateral flow immunochromatography rapid test (Qingdao Hightop Biotech Co., Ltd), using a capture method for the qualitative detection of SARS-CoV-2 IgM/IgG antibodies following the manufacturer's instructions (IgM sensitivity: 82%, specificity: 96%; IgG sensitivity: 93%, specificity: 98%) [8]. Plasma samples were thawed, and 10 µl were transferred to the Hightop test cassette. Subsequently, two drops (80–100µl) of chromatography buffer were added on top of the serum in the test cassette and incubated at room temperature for 15–20 minutes. The test cassettes were observed for the presence or absence of three bands, one corresponding to the internal control, one corresponding to anti-SARS-CoV-2 IgG, and one corresponding to anti-SARS-CoV-2 IgM. Each test required the

control band to be observed. A result was considered invalid if the development of bands took more than 20 minutes or when no band was observed for the internal control.

## Data analysis

All data used for this report can be found at Mendeley data (doi:10.17632/h43mdmz6vv.1). Given the recent surge of SARS-CoV-2 cases, the total seroprevalence was estimated by pooling IgG and IgM positive samples. Seroprevalence by age was calculated as the sum of the percentage of the total seropositivity within the age group divided by the total number of children sampled within the same age group. To quantify the relationship between sex, age, and total population (adults and children) distribution among houses inhabited by sampled children and, total seroprevalence (binary variable: positive = 1, negative = 0), generalized linear mixed models with a binomial link function and a random intercept associated with each house ID were implemented. To explore the relationship between house crowding and total seropositivity, we use a crowding idex measured as the total number or ocupants per household reported by the National Institute of Statistics and Geography (INEGI) and adjusted for our own data. INEGI resported 3.3 occupants per house in the last 2020 population census [9] while we calculated an average of 5 people per house for our study. Based on this information we split households into two groups (houses with more than five inhabitants versus houses with five or less inhabitants) and compared seroprevalences using a Chi-squared test.

Seroprevalence to SARS-CoV-2 was also calculated by geographic cluster (N = 50 clusters measuring 5x5 city blocks each) and analyzed using global and local spatial statistics. A weighted K-function, $L(d)$ quantified the global aggregation of exposure in children of Mérida [10]. Briefly, L(d) utilizes the centroid of the 5x5 city block clusters to quantify whether there is a tendency of clusters of high seroprevalence to be near other clusters of high seroprevalence. The observed $L(d)$ was contrasted with the expectation of complete spatial randomness after conducting 999 Monte Carlo simulations up to a distance separating clusters of 10 km. Local aggregation was quantified with the **Getis-Ord $G_i^*(d)$** hotspot analysis to identify which 5x5 city-block clusters had neighbors with high seroprevalence, compatible with a hotspot of SARS-CoV-2 transmission within the city [11]. As before, 999 Monte Carlo simulations were conducted and compared the observed value of $G_i^*(d)$ with the expectation of complete spatial randomness using increasing distances up to 10 km. The distance up to which the *P*-value of $G_i^*(d)$ was lower than 0.05 was plotted as a buffer in a map of the city of Mérida containing the seroprevalence of SARS-CoV-2 in children.

All data management and analyses were performed using Stata 17.0 (StataCorp, Tx) and R programming environment (https://www.r-project.ct.org).

## Results

### Identification and enrollment of study cohort

We enrolled an ambulatory population of 1,005 children (2–15 years old; median = 9 [CI: 6–12] years old) from 706 households in the city of Mérida, Yucatán, México. In an attempt to obtain a representative cohort, participants were recruited from randomly selected regions throughout the city (**Fig 1**) and all children within the designated areas were eligible to participate. Male and female children of between the ages of two and fifteen years old were enrolled at an equivalent rate (**Fig 2**). From May to June of 2021, teams of nurses and social workers visited each participant at their home and took a sample of veinous blood from each. This blood was transported by coolers to a central laboratory for processing. The serum was extracted from each sample and tested by commercially available lateral flow chromatography kits for the presence of anti-SARS-CoV-2 IgM and IgG antibodies.

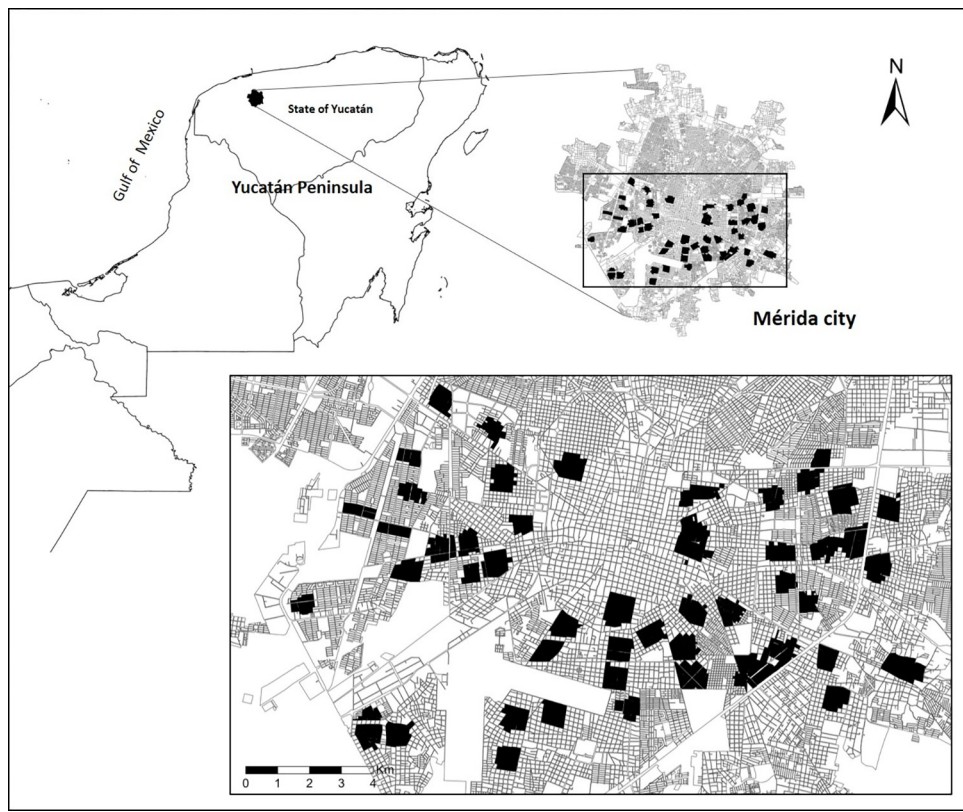

**Fig 1. Working area, showing the spatial distribution of 50 neighbor clusters in the city of Mérida, Yucatán.** Inset shows the location of Mérida within Yucatán State and México. Base layers were downloaded from México's census database (https://www.inegi.org.mx/app/mapa/espacioydatos/default.aspx).

## Seroprevalence of anti-SARS-CoV-2 antibodies in the cohort

Any participant with SARS-CoV-2 specific antibodies in their serum sample was defined as seropositive. We quantified a total seroprevalence of 25.5% among all children (**Table 1**), with seroprevalence not differing statistically among ages or sexes (**Fig 3**). Because the prevalence of exposure to endemic pathogens should increase with age, these results were consistent with the introduction of a novel pathogen. Additionally, 2.6% of the asymptomatic children tested positive for IgM, indicating a very recent exposure to SARS-CoV-2 (**Table 1**).

To determine if anti-SARS-CoV-2 antibodies were more prevalent in any geographical region of the city, a spatial distribution analysis of the positive samples was performed. While heterogeneity in the distribution of seroprevalence was observed (**Fig 4A**), the global weighted K-function detected no evidence of aggregation of seroprevalence at any of the distances evaluated (**Fig 4B**). These results suggest that, on average, high seroprevalence clusters were not close to each other. When testing for local aggregation (whether any clusters showed evidence of aggregation of high seroprevalence with their neighbors) two hotspots (measuring 2.5 and 5.0 km) of high seroprevalence were detected in the west of Mérida (**Fig 4C**).

## Risk factors for the presence of anti-SARS-CoV-2 antibodies

Because of the rapid introduction and easy transmission of SARS-CoV-2 in Mérida and elsewhere, we hypothesized that demographic factors would have a modest effect on the previous exposure. Linear regression analyses, comparing seroprevalence with age and gender, showed

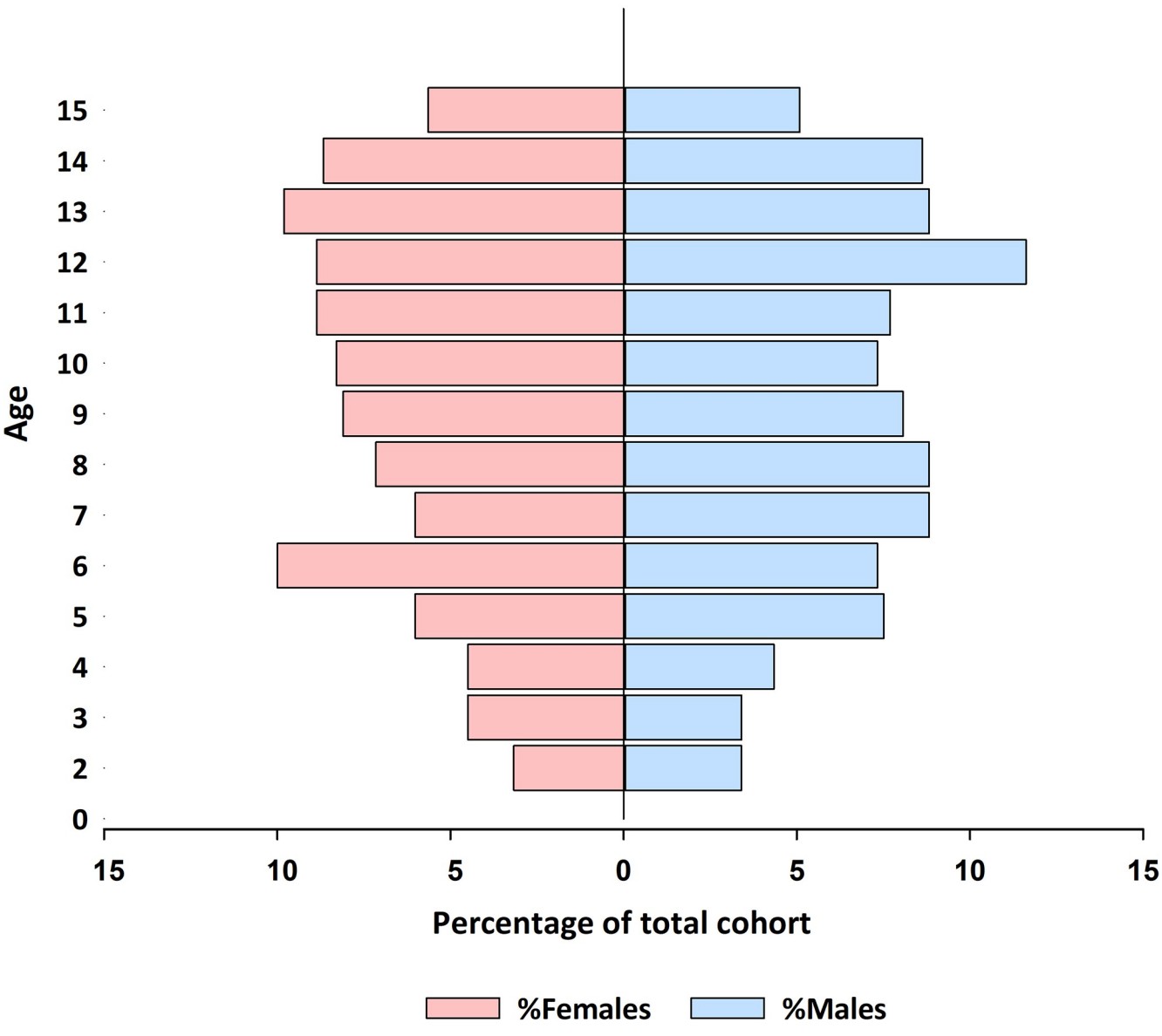

**Fig 2. Population age structure of children (2–15 years old) participating in the study.** The age of children in the cohort at enrolment and sampling time in Mérida, Yucatán, México. Data are presented as a percentage of the total cohort.

**Table 1. Seropositivity of children enrolled in the study.**

|  |  | IgM | | Total |
|---|---|---|---|---|
|  |  | **Negative** | **Positive** |  |
| *IgG* | **Negative** | 749 (74.5%) | 6 (0.6%) | 755 (75.1%) |
|  | **Positive** | 230 (22.9%) | 20 (2.0%) | 250 (24.9%) |
|  | **Total** | 979 (97.4%) | 26 (2.6%) | 1,005 |

The number and percentage of enrolled children (2–15 years old) that tested negative or positive for anti-SARS-CoV-2 IgG and/or IgM. The total seropositivity (IgG + IgM) was 25.5% (256/1,005).

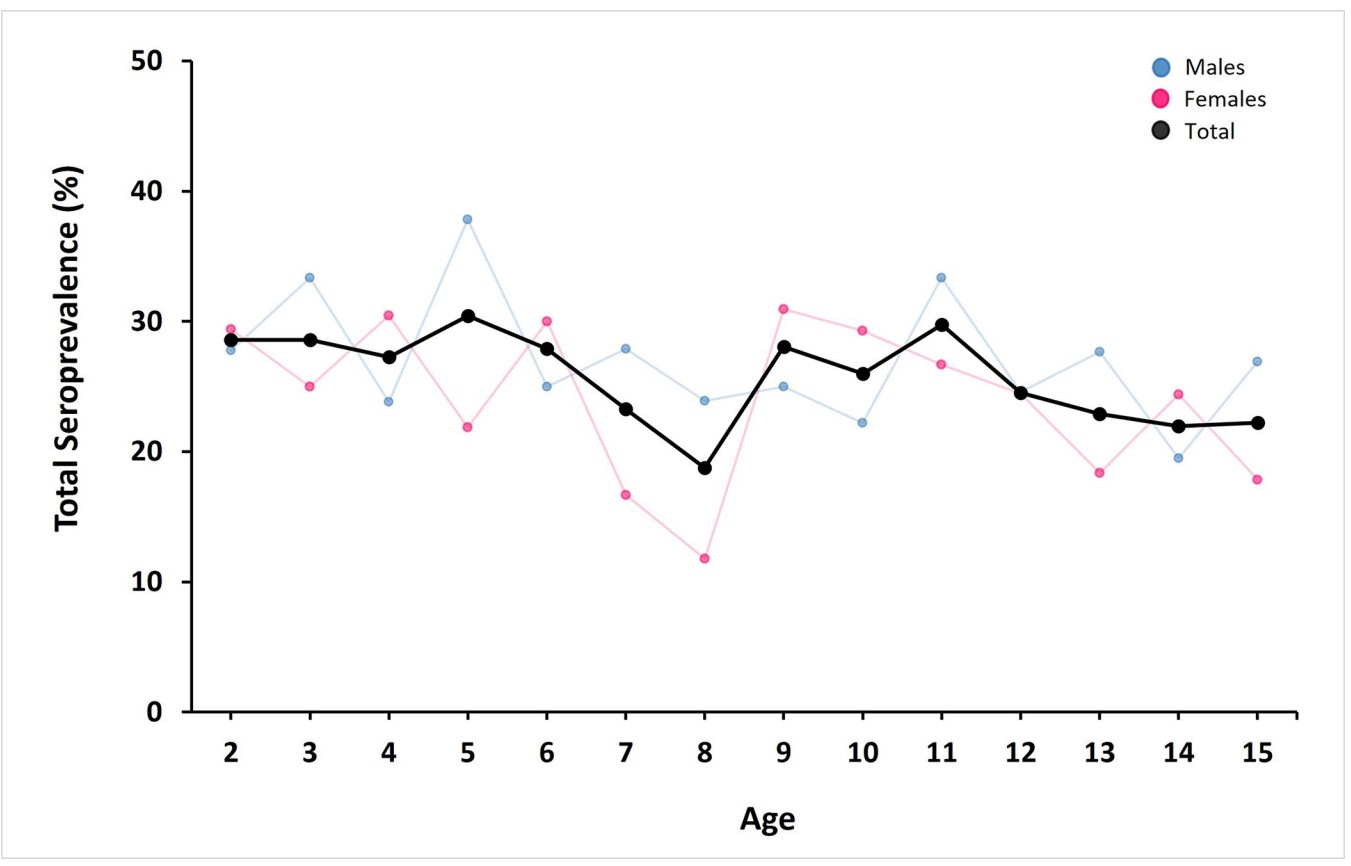

**Fig 3. Seroprevalence of SARS-CoV-2 antibodies in children by age, sex and total.** Enrolled children (2–15 years old) had no previous report of COVID-19 symptoms. Seroprevalence of anti-SARS-CoV-2 antibodies is presented in males (blue), females (red) and the total cohort (black). Samples were collected from March to June 2021 in Mérida, Yucatán, México.

that male children are slightly more at risk (Odds ratio = 1.17), however, this trend was not statistically significant (p = 0.54) (**Table 2**). Similarly, when broken down by age, no effect on the likelihood of previous exposure to SARS-CoV-2 was found (Odds ratio = 0.99). Furthermore, no sociologic or demographic factors were found to significantly affect seroprevalence save household size (Table 2). These results suggest that demography does not affect the risk of previous exposure to SARS-CoV-2 in this cohort.

## Household size as a risk factor for anti-SARS-CoV2 antibodies

Until the time of our study, the city of Mérida relied primarily on an intensive set of non-pharmacological interventions to limit the transmission of SARS-CoV-2, including social distancing, school closures, and a curfew to decrease the mobility of people at night. Because of this limited movement, we hypothesized most children were likely exposed to SARS-CoV-2 in their homes. We matched the serology results of each participant with their household size, including adults and children, and performed statistical analyses. We found that children living in homes with more than five inhabitants, the average number of inhabitants of a household in our trial, were 10.2% ($X^2_{(df = 1)}$ = 9.96, P <0.001) more likely to have been previously exposed to the SARS-CoV-2 than children living in houses with five or less people (**Fig 5**). Indeed, we calculated a positive Odds-Ratio for SARS-CoV-2 exposure of 1.27 (95% CI: 1.06–1.47) for every additional member in a household (**Table 2**).

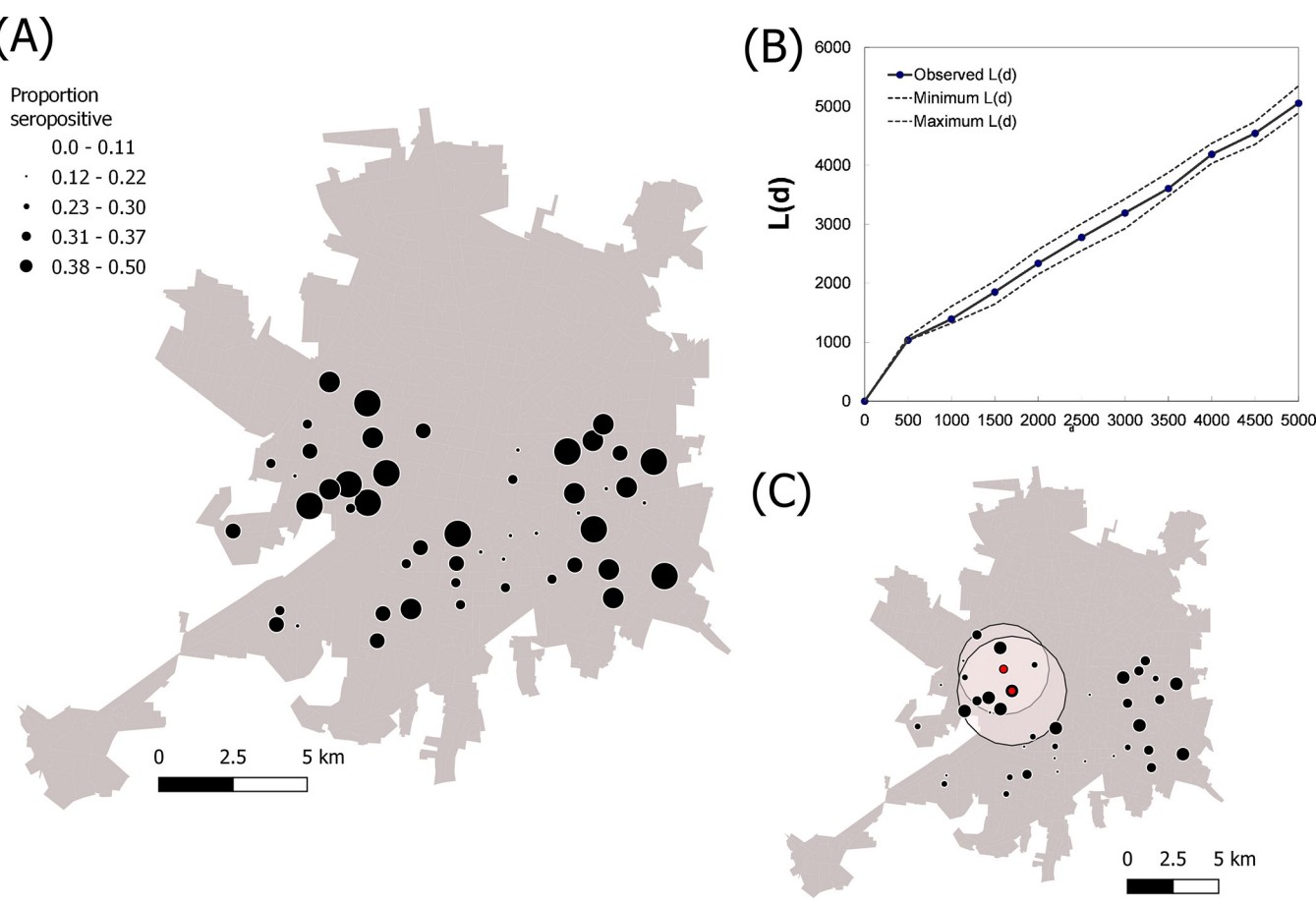

**Fig 4. Spatial pattern of SARS-CoV-2 seroprevalence in children of Mérida, Yucatán, México.** (**A**) Distribution of average seroprevalence in children, distributed across 50 clusters measuring 5x5 city blocks (and mapped to the cluster's centroid). (**B**) Results of global weighted K-function L(d) quantifying the average aggregation of high seroprevalence of SARS-CoV-2 at increasing distances from each study cluster. The plot shows the observed L(d) and its comparison with the 95% confidence interval of the expectation of complete spatial randomness. (**C**) Results from a local **Getis G$_i$\***(**d**) hotspot analysis showing the clusters identified as hotspots (in red) and the extent of each hotspot (pink buffers). Base layers were downloaded from México's census database (https://www.inegi.org.mx/app/mapa/espacioydatos/default.aspx).

## Discussion

Our study quantified significant levels of exposure to SARS-CoV-2 in the pediatric population of a large Mexican city during 61 days from February to June of 2021. Even with this large,

**Table 2. Relationship between total seroprevalence (IgG+IgM) and demographic and household factors.**

| Characteristic | OR | 95% CI | P value |
| --- | --- | --- | --- |
| *Sex* | | | |
| Female | Ref. | -- | -- |
| Male | 1.17 | 0.71–1·90 | 0.54 |
| *Age* | 0.99 | 0.92–1.07 | 0.77 |
| **Household Size** | 1.27 | 1.06–1.47 | 0.008 |

Results from a binomial generalized linear mixed model testing the association between IgG/IgM seropositivity and each factor. The Odds-Ratio (OR) was calculated for each independent covariate, with males being reported in reference (Ref.) to females.

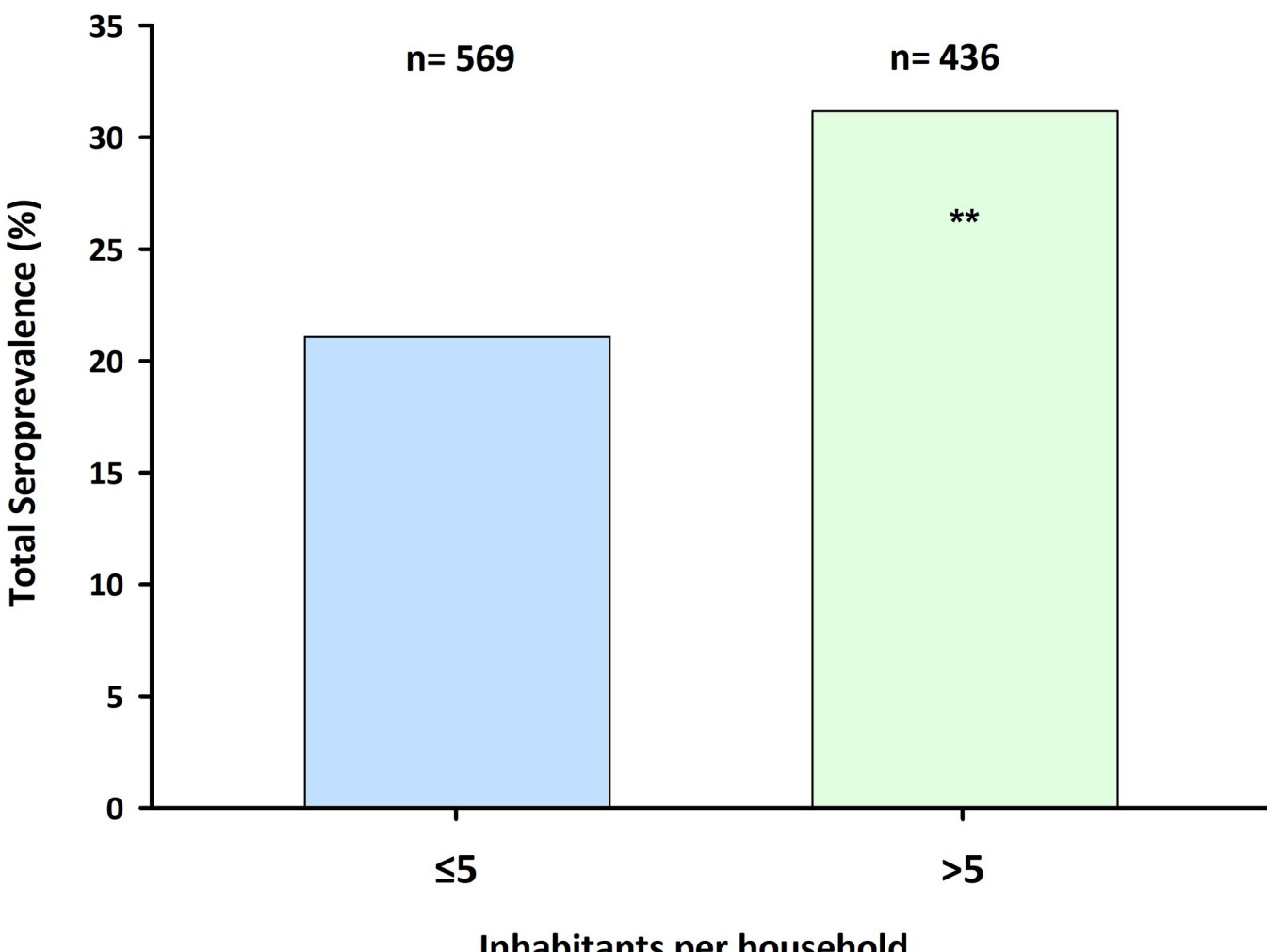

**Fig 5. Seroprevalence of anti-SARS-CoV-2 antibodies in children by household size.** The seroprevalence for anti-SARS-CoV-2 antibodies in houses inhabited by five or less (blue) total people (adults and children) or more than five (green) people collected from March to June 2021 in Mérida, Yucatán, México. A significantly higher seroprevalence was found in the >5 group (31.2%) compared to the ≤ 5 group (21.0%; $X^2_{(df=1)}$ = 9.96, p-value <0.001).

exposed population, COVID-19 vaccine doses have been distributed in Mérida only to the adult and high-risk population. As of October 2021, most schools in México have resumed in-person classes following social distancing, mask-wearing policies, and optional attendance in some schools. Based on Mérida's 2020 census reporting of 185,846 children <15 years old, we infer that there is a population of 138,455 SARS-CoV-2 susceptible children starting in-person school in the city [9]. This silent majority of unvaccinated children may act as a reservoir of infection that can lead to the generation of viral variants and source of outbreaks, with potential consequences for future transmission waves and vaccine efficacy.

We observed that SARS-CoV-2 seroprevalence in children was not affected by demographic factors such as age and sex but was affected by household size. The demographic results are consistent with the introduction of a novel infectious agent with no previous serum immunity present within the population. If a virus is endemic or persistently re-introduced, one would expect to observe an increased seroprevalence in older individuals due to increasing exposure lengths [12]. Furthermore, the presence of closely-related pathogens that can induce baseline

cross-reactive antibodies might also be observed in cross-sectional studies. The only socio-economic factor observed to affect seroprevalence was household size. Children from households with more adults and/or children were more likely to have been previously exposed to SARS-CoV-2. Combined with the local and regional travel restrictions imposed in the Yucatán state in 2020–2021, these results suggest that the most critical factor for exposure to SARS-CoV-2 is simply the number of people that the subject interacts with. Spatial analyses revealed a moderate level of local clustering in the West of the city, with cluster sizes ranging between 2.5–3.0 km. Such findings point to the occurrence of the differential intensity of pediatric SARS-CoV-2 transmission throughout the city, as described for passive surveillance case data for adults in multiple cities in China [13] as well as New York City and Chicago [13,14]. These results have profound implications as travel restrictions are eased and especially as schools are opened. The exposure rate of unvaccinated and serologically naïve children will very likely increase by a large percentage. This will likely lead to an increased exposure rate in these children and could potentially lead to enhanced transmission rates within the community.

Unvaccinated individuals pose a risk to populations as they can act as reservoirs for SARS-CoV-2 infection. Children present an even more significant risk as they are unlikely to exhibit symptoms of Covid-19 and are not tested at the same rates as adults. In 2021, the world has witnessed the emergence of several SARS-CoV-2 variants that pose a serious threat to virus control efforts. One important example is the SARS-CoV-2-B.1.617.2 "Delta" variant which emerged in India in December of 2020 [15]. This virus proved to have a higher reproductive rate than previous variants and could escape antibodies produced in response to infections by other SARS-CoV-2 viruses [16,17]. While currently available vaccines can control the Delta variant, the efficacy against this variant is reduced [18]. The ability of the SARS-CoV-2 virus to escape currently available vaccines through mutation of the viral Spike protein is a significant threat to global public health. These mutations are more likely to arise among undervaccinated populations that can be infected, provide an environment conducive to the rise of variants, and spread these variants among their close peers. As children return to school, the virus may be presented with the optimal conditions to produce vaccine escape mutants. Thus, monitoring the prevalence of SARS-CoV-2 infection in unvaccinated children should be a priority for public health officials as vaccines are distributed.

The results of this study describe the seroprevalence of anti-SARS-CoV-2 antibodies in an ambulatory population of children in a Latin American country. The seroprevalence in children in Mérida was relatively higher than that estimated in studies in the United States, United Kingdom, and France, and more in line with the estimates for the outbreak study in Austria [19–22]. This study benefited from a large population of untested child participants and the many social and georgraphic variables considered, but the study was limited by being a cross-sectional instead of longitudinal study of SARS-CoV-2 reactivity that was only performed in one city in the Yucatán peninsula. Furthermore, due to the binomial nature of the commercial kit used for this study, differential antibody titers in the population were not measured and thus any differences in the magnitude of the antibody response is not known. Further work is needed to determine if these results are applicable to countries in Central and South America. Furthermore, an analysis of children not living in an urban setting may be necessary. Children in rural areas are less likely to be vaccinated as quickly as those in urban areas [23]. Thus, the seroprevalence in these populations may be an important factor for preventing the unfavorable outcomes described above. It is important that children be vaccinated as soon as possible, but until that happens, it is also essential to be aware of SARS-CoV-2 spread within child-aged populations.

## Supporting information

**S1 Text. Inclusivity in global research questionnaire.**
(DOCX)

## Acknowledgments

We thank the nurses, doctors, social workers, and database staff of the TIRS project for collecting these data. The opinions expressed in this article are those of the authors and do not reflect the view of the National Institutes of Health, the Department of Health and Human Services, or the United States government.

## Author Contributions

**Conceptualization:** Guadalupe Ayora-Talavera, Oscar D. Kirstein, Henry Puerta-Guardo, Pablo Manrique-Saide, Gonzalo M. Vazquez-Prokopec, James T. Earnest.

**Data curation:** Guadalupe Ayora-Talavera, Oscar D. Kirstein, Henry Puerta-Guardo, Carlos Culquichicon, Gonzalo M. Vazquez-Prokopec, James T. Earnest.

**Formal analysis:** Guadalupe Ayora-Talavera, Oscar D. Kirstein, Henry Puerta-Guardo, Carlos Culquichicon, Gonzalo M. Vazquez-Prokopec, James T. Earnest.

**Funding acquisition:** Gonzalo M. Vazquez-Prokopec.

**Investigation:** Guadalupe Ayora-Talavera, Oscar D. Kirstein, Henry Puerta-Guardo, Gloria A. Barrera-Fuentes, Desiree Ortegòn-Abud, Manuel Parra, Fernando Peña-Miranda, Carlos Culquichicon, James T. Earnest.

**Methodology:** Guadalupe Ayora-Talavera, Oscar D. Kirstein, Henry Puerta-Guardo, Manuel Parra, Fernando Peña-Miranda, Norma Pavia-Ruz, Afshin Beheshti, Nídia S. Trovão, Pilar Granja-Pérez, Pablo Manrique-Saide, Gonzalo M. Vazquez-Prokopec, James T. Earnest.

**Project administration:** Guadalupe Ayora-Talavera, Oscar D. Kirstein, Henry Puerta-Guardo, Gloria A. Barrera-Fuentes, Desiree Ortegòn-Abud, Azael Che-Mendoza, Norma Pavia-Ruz, Pilar Granja-Pérez, Pablo Manrique-Saide, Gonzalo M. Vazquez-Prokopec, James T. Earnest.

**Resources:** Afshin Beheshti, Nídia S. Trovão.

**Supervision:** Guadalupe Ayora-Talavera, Oscar D. Kirstein, Henry Puerta-Guardo, Gloria A. Barrera-Fuentes, Desiree Ortegòn-Abud, Azael Che-Mendoza, Norma Pavia-Ruz, Pilar Granja-Pérez, Pablo Manrique-Saide, Gonzalo M. Vazquez-Prokopec, James T. Earnest.

**Writing – original draft:** Guadalupe Ayora-Talavera, Oscar D. Kirstein, Gonzalo M. Vazquez-Prokopec, James T. Earnest.

**Writing – review & editing:** Guadalupe Ayora-Talavera, Oscar D. Kirstein, Henry Puerta-Guardo, Gloria A. Barrera-Fuentes, Desiree Ortegòn-Abud, Azael Che-Mendoza, Manuel Parra, Fernando Peña-Miranda, Carlos Culquichicon, Norma Pavia-Ruz, Afshin Beheshti, Nídia S. Trovão, Pilar Granja-Pérez, Pablo Manrique-Saide, Gonzalo M. Vazquez-Prokopec, James T. Earnest.

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
