## [Decision Letter · Decision Letter 0]

21 Feb 2022

PGPH-D-21-01015

SARS-CoV-2 antibody prevalence in a pediatric cohort of unvaccinated children in Mérida, Yucatán, México

Dear Dr. Earnest,

Thank you for submitting your manuscript to PLOS Global Public Health. After careful consideration, we feel that it has merit but does not fully meet PLOS Global Public Health’s publication criteria as it currently stands. Therefore, we invite you to submit a revised version of the manuscript that addresses the points raised during the review process.

We look forward to receiving your revised manuscript.

Kind regards,

Dr. Hossam M Ashour, Ph.D.

Academic Editor

Journal Requirements:

1. Please include a complete copy of PLOS’ questionnaire on inclusivity in global research in your revised manuscript. Our policy for research in this area aims to improve transparency in the reporting of research performed outside of researchers’ own country or community. The policy applies to researchers who have travelled to a different country to conduct research, research with Indigenous populations or their lands, and research on cultural artefacts. The questionnaire can also be requested at the journal’s discretion for any other submissions, even if these conditions are not met.  Please find more information on the policy and a link to download a blank copy of the questionnaire here: https://journals.plos.org/plosone/s/best-practices-in-research-reporting. Please upload a completed version of your questionnaire as Supporting Information when you resubmit your manuscript.

2. In the online submission form, you indicated that "All data that support the findings of this work are available, upon request, from JTE.". All PLOS journals now require all data underlying the findings described in their manuscript to be freely available to other researchers, either 1. In a public repository, 2. Within the manuscript itself, or 3. Uploaded as supplementary information.

3. Please provide us with a direct link to the base layer of the map used in Figs 1 and 4 and ensure this location is also included in the figure legend. 

Please note that, because all PLOS articles are published under a CC BY license (creativecommons.org/licenses/by/4.0/), we cannot publish proprietary maps such as Google Maps, Mapquest or other copyrighted maps. If your map was obtained from a copyrighted source please amend the figure so that the base map used is from an openly available source.

Please note that only the following CC BY licences are compatible with PLOS licence: CC BY 4.0, CC BY 2.0  and CC BY 3.0, meanwhile such licences as CC BY-ND 3.0 and others are not compatible due to additional restrictions. If you are unsure whether you can use a map or not, please do reach out and we will be able to help you. 

The following websites are good examples of where you can source open access or public domain maps:

Additional Editor Comments (if provided): Please make sure to address all reviewers' comments.

Reviewers' comments:

Reviewer's Responses to Questions

**Comments to the Author**

1. Does this manuscript meet PLOS Global Public Health’s publication criteria? Is the manuscript technically sound, and do the data support the conclusions? The manuscript must describe methodologically and ethically rigorous research with conclusions that are appropriately drawn based on the data presented.

Reviewer #1: Yes

Reviewer #2: Yes

Reviewer #3: Partly

Reviewer #4: Partly

2. Has the statistical analysis been performed appropriately and rigorously?

Reviewer #1: No

Reviewer #2: Yes

Reviewer #3: I don't know

Reviewer #4: Yes

3. Have the authors made all data underlying the findings in their manuscript fully available (please refer to the Data Availability Statement at the start of the manuscript PDF file)?

Reviewer #1: Yes

Reviewer #2: No

Reviewer #3: Yes

Reviewer #4: Yes

4. Is the manuscript presented in an intelligible fashion and written in standard English?

Reviewer #1: Yes

Reviewer #2: Yes

Reviewer #3: Yes

Reviewer #4: Yes

5. Review Comments to the Author

Reviewer #1: I indicated "No" in response to the question "Has the statistical analysis been performed appropriately and rigorously". This is not because the analyses that the authors conducted are flawed, but because there are three testable factors (that I can identify) which could have led to spurious conclusions, but which were not tested. Once the authors explore these questions, the statistical analysis should be deemed appropriate and rigorous:

1. Is there any correlation between the arm of the RCT in which children are enrolled and SARS-CoV-2 antibody prevalence? Could this have any impact on the conclusions?

2. What are the rates of non-response or non-consent to the socio-demographic survey and blood draws? Could this have any impact on the conclusions?

3. Given the rapidly-changing incidence rates and relatively large wave of infection that happened in Mexico in January of 2021, I assume that the clusters which were surveyed later in the study have higher seroprevalence than the clusters which were surveyed early, unless this study somehow surveyed households as a simple random sample. Please: a) Confirm whether the survey protocol sampled whole clusters in serial or whether the survey protocol was able to approximate a random sample of households over time, and b) Report whether/how the statistical analyses (the age/sex risk factor analysis, the household size correlation analysis and the K-function spatial analysis) accounted for temporal confounding in drawing their conclusions. For example, could it be possible that the spatial analysis found no tendency for neighboring clusters to share high seroprevalence because of the order in which the clusters were sampled? Could it be possible that household size was found to correlate with seroprevalence because clusters tended to have larger household size later in the surveying protocol?

In addition, I have three suggestions that I believe will enhance the quality and significance of the manuscript:

1. Please include the complete socio-demographic survey instrument used in this study, if not restricted by original IRB. Why were age, sex and household size the only determinants explored?

2. What is the justification for testing a binary variable (>5 inhabitants vs <=5 inhabitant) instead of a continuous household size variable? The linear odds ratio presented in Table 2 seems sufficient.

3. I believe there is a discrepancy in the definition of a large household size as described in the Methods and Results sections. The Methods section states "houses with more than five inhabitants" and the Results section states "homes with more than six inhabitants". Were two definitions tested or should one of these sentences be corrected?

Reviewer #2: Overall comment

The authors have touched upon a very important topic. There is limited evidence about pediatric infection with COVID-19 relative to the massive evidence that has been generated about the disease in adults since the onset of the pandemic. The authors have also used very robust statistical and analytical techniques to analyze the data and have presented it well.

Introduction

The ending paragraph of the introduction need not carry the details of the methodology or study setting as the details are mentioned in the methods section.

Methods

Study design and sample size

While the authors have done well to mention that the samples that were analyzed for the study were part of a larger on-going trial and were randomly selected, the method followed for random selection of samples has not been specified.

If the original study was using a cluster randomization, how was this taken into account when selecting the samples for this study?

What were the parameters or underlying assumptions for the sample size?

The eligibility criteria for selection of participants have not been stated clearly either. Though it is mentioned that none of the participants had a prior reported COVID-19 infection, it is unclear if that had any basis for study eligibility? Lines 85-86 (page 4; Introduction) and lines 121-122 (Page 5; Methods) seem like contradictory statements and should be revised. If the aim of the study was to estimate the prevalence of COVID-19 among children through measurement of antibodies, then excluding those with a history of prior infection would lead to an underestimation.

Also, the participants of this study were part of a trial and may have been selected through an eligibility criteria that set them apart from the general population. How would this bias the result or what impact would it have on the estimates?

Also, if more than one eligible child was present in the household, were all selected?

A lot of repetition about the enrollment and sample collection throughout the text. It is unclear what information was collected at the time of enrollment. There was a considerable lag between enrollment period and period of sample collection. Was the information on prior history of COVID-19 collected at the time of enrollment or time of sample collection?

How long after collection were the samples analyzed for the presence of anti-SARS-CoV-2 antibodies?

Analysis

The abstract states that the socioeconomic and demographic factors were also studied. Apart from household size, there is no mention of any other socioeconomic measure in the analysis. If other measures were examined and no difference was observed, it should be mentioned in the results.

The authors could have also collected information regarding crowding index that takes into account the number of living spaces in the house/dwelling for the given number of occupants. That may be a better predictor of household transmission than just the household size. It also gives an estimate of any difference due to socio-economic status.

Table 2 footnote should also indicate the reference category used for household size

Results

There is too much discussion in the results section. Some details mentioned in the results would be more appropriate in the discussion section.

Discussion

Study strengths and limitations have not been discussed. It would be helpful to know the possible impact that they can have on the estimates obtained in the study.

Reviewer #3: It is a study that provides valuable information to understand the dynamics of SARS-CoV-2 transmission in children. However, it is necessary to clarify the methodology, beginning with the study's design. The authors declare it is a cohort study, but it seems that there is no follow-up. It looks like a cross-sectional study nested in a longitudinal design (the TIRS trial); please clarify, and follow the STROBE statement to describe the central aspect of the methods.

It is unclear how the authors accessed the children and their blood samples. In line 118, they say, "We selected a subset of 1,005 serum samples of children for anti- SARS-CoV-2 antibody testing". But in line 125, they explain, "Field personnel collected blood specimens by house-to-house visits." These two phrases seem contradictory.

It is unclear if they included all the children who participated in the Targeted Indoor Residual Spraying (TIRS) trial in Mérida, and how that study enrolled the children. Were there rejections? It is unusual not to have it when requesting venipuncture for children.

Please provide the rational to include subjects who reported no previous or current symptoms of SARS-CoV-2 infection.

Finally, this is not the first Latin American study to include children, although it is probably the first with young children.

Reviewer #4: I do appreciate Guadalupe et al… work. The authors have conducted large analysis incorporating from several samples. However, corrections on introduction, Methods, result and Conclusion should be addressed well. In addition to this, grammar, editing and following the guideline on the tables and figures need to be addressed.

Abstract section

On the last part, it be better if you recommend based on your pertinent finding instead of saying "This is the first analysis of a large cohort of children in a Latin American country and the results may inform public health decisions …………" which seems like more general.

List the abbreviations of SARS, IgG and IgM (when you use them in the first time)

Introduction

Paragraph one of this section is very interesting, but the line 54 of the paragraph one be best if you begin with definition of your outcome variable.

Line 70: said that "Here, we provide insight on the prevalence…….." avoid the unnecessary words such us we, I and etc. throughout in the document.

The last paragraph of the introduction section [from line 81-88] should rewrite it again. This is better to be in the result and the methods section.

General comment: The introduction is good but it need to address the severity, consequences and clear gap of the problem in this section.

Method section

The study site is not clearly addressed for readers (the full address of the Mérida)

Data collection procedure, sample size determination, study populations, exclusion criteria's, data collectors, Measurements, data quality controls Need to be addressed well in this section.

Result Section

The narrative of the biographic data in this section need to address clearly

Follow the guide lines of the Journal for the all the tables and figures.

Discussion

In the paragraph two (line 306) you discussed the finding of the study from other but the study areas of the other studies are not listed (even though it is cited as 13 and 14).

There is no conclusion, limitation (optional) and Recommendation in this paper. It is better to include conclusion (inclusively recommendation under this).

Funding issue, Abbreviation and Acronyms are not incorporated in this paper

6. PLOS authors have the option to publish the peer review history of their article (what does this mean?). If published, this will include your full peer review and any attached files.

**Do you want your identity to be public for this peer review?** For information about this choice, including consent withdrawal, please see our Privacy Policy.

Reviewer #1: **Yes: **David E Phillips

Reviewer #2: No

Reviewer #3: No

Reviewer #4: No

---

## [Decision Letter · Decision Letter 1]

25 May 2022

SARS-CoV-2 antibody prevalence in a pediatric cohort of unvaccinated children in Mérida, Yucatán, México

PGPH-D-21-01015R1

Dear Dr. Earnest,

We are pleased to inform you that your manuscript 'SARS-CoV-2 antibody prevalence in a pediatric cohort of unvaccinated children in Mérida, Yucatán, México' has been provisionally accepted for publication in PLOS Global Public Health.

Best regards,

Hossam M Ashour, Ph.D.

Academic Editor

Reviewer Comments (if any, and for reference):

Reviewer's Responses to Questions

**Comments to the Author**

1. If the authors have adequately addressed your comments raised in a previous round of review and you feel that this manuscript is now acceptable for publication, you may indicate that here to bypass the “Comments to the Author” section, enter your conflict of interest statement in the “Confidential to Editor” section, and submit your "Accept" recommendation.

Reviewer #1: All comments have been addressed

Reviewer #2: All comments have been addressed

Reviewer #4: All comments have been addressed

2. Does this manuscript meet PLOS Global Public Health’s publication criteria? Is the manuscript technically sound, and do the data support the conclusions? The manuscript must describe methodologically and ethically rigorous research with conclusions that are appropriately drawn based on the data presented.

Reviewer #1: Yes

Reviewer #2: Yes

Reviewer #4: Yes

3. Has the statistical analysis been performed appropriately and rigorously?

Reviewer #1: Yes

Reviewer #2: Yes

Reviewer #4: Yes

4. Have the authors made all data underlying the findings in their manuscript fully available (please refer to the Data Availability Statement at the start of the manuscript PDF file)?

Reviewer #1: Yes

Reviewer #2: Yes

Reviewer #4: Yes

5. Is the manuscript presented in an intelligible fashion and written in standard English?

Reviewer #1: Yes

Reviewer #2: Yes

Reviewer #4: Yes

6. Review Comments to the Author

Reviewer #1: No further comments

Reviewer #2: Commendable effort has been made by the authors in revising the manuscript and detailed responses have been given for each reviewer comments. However, this reviewer feels that adding GPS coordinates of the study site may not be necessary and is not common practice

Reviewer #4: The Authors addressed the comments very well except for some editorial errors. I recommend the Author edit throughout the paper.

7. PLOS authors have the option to publish the peer review history of their article (what does this mean?). If published, this will include your full peer review and any attached files.

**Do you want your identity to be public for this peer review?** For information about this choice, including consent withdrawal, please see our Privacy Policy.

Reviewer #1: **Yes: **David E Phillips

Reviewer #2: **Yes: **Nadia Ansari

Reviewer #4: No
